# Performance-Based Assessment of Bridges with Novel SMA-Washer-Based Self-Centering Rocking Piers

**DOI:** 10.3390/ma15196589

**Published:** 2022-09-22

**Authors:** Jiawei Chen, Dong Liang, Xin You, Hao Liang

**Affiliations:** 1College of Civil Engineering, Tongji University, Shanghai 200092, China; 2College of Civil Engineering, Qinghai University, Xining 810016, China; 3College of Civil Engineering, Southeast Jiaotong University, Chengdu 611756, China

**Keywords:** self-centering rocking (SCR) piers, shape memory alloy (SMA), seismic fragility, resilience, life-cycle loss

## Abstract

This study discussed a novel self-centering rocking (SCR) bridge system equipped with shape memory alloy (SMA)-based piers, with a particular focus on the benefit of the SCR bridge system in a life-cycle context. The study commences with an introduction of the SCR bridge system; subsequently, a life-cycle loss and resilience assessment framework for the SCR bridge system is presented. Specifically, the seismic fragility, resilience, and life-cycle loss associated with the SCR and conventional bridge systems were addressed. The proposed life-cycle assessment framework was finally applied to two highway bridges with and without SMA washer-based rocking piers, considering the representative hazard scenarios that could happen within the investigated regions. The results revealed that the novel SCR pier bridge system slightly increased the bearing displacement but extensively reduced the pier curvature ductility due to the rocking mechanism. The SCR bridge system kept a lower life-cycle loss level and exhibited more resilient performance than the conventional bridge, especially in the region with higher seismic intensities. Indirect loss can be significantly larger than the direct loss, specifically for the earthquakes with a relatively low probability of occurrence. The SCR bridge system outperformed the conventional system in terms of recovery time, where a quick recovery after an earthquake and drastically decreased the social and economic losses.

## 1. Introduction

Although ductility-based seismic design philosophy has been employed for a long time, severe damage to bridges (e.g., unseating of girders) in recent earthquakes (e.g., Maduo earthquake, China, 2021) indicates its limitation. Overly large residual deformation may compromise the normal function of the bridges after earthquakes, and this issue is gaining increasing attention. To ensure normal operation of the lifeline systems, new design principles associated with residual deformation in seismic codes have been appended in many countries (e.g., US, Japan, and New Zealand) [1]. For instance, rocking bridge piers have been gaining attraction because of their small residual deformation property [2] and fast precast construction characteristic [3,4,5,6]. The objective of rocking is to remarkably decrease the input energy due to earthquakes by elongating the fundamental period of vibration. Some experimental studies [7] have been successfully carried out to verify the concept. The design allows the piers to rock around the foundation/footing, resulting in the alleviation of seismic damage. In order to avoid overturning of rocking bridges, typical post-tensioned (PT) rocking piers have been proposed together with test verification [8,9,10,11]. To further improve the performance of the rocking piers subjected to severe earthquakes, a series of novel supplementary self-centering and/or energy dissipation devices have been developed and examined [12,13,14,15,16].

Although combining PT tendons with energy dissipaters can remarkably decrease structural damage and residual deformation, repair or replacement of the energy dissipaters after earthquakes is time consuming and costly. Corrosion may also be an issue for metal energy dissipaters. In this regard, shape memory alloy (SMA), a novel class of metal, has been recently employed in bridge structures [17,18,19,20,21,22] as well as other engineering structural systems [23,24,25,26,27,28,29,30,31,32,33,34] to enhance their seismic resilience. At the austenite phase, SMA can exhibit superelasticity at room temperature, and is capable of recovering large strains (up to 8~10% strain) after experiencing earthquake excitation. A representative investigation was conducted by Varela and Saiidi [20], who examined the feasibility of using SMA bars at the plastic hinge zone of RC piers. The results indicated that except for the bucking of the SMA bars, the RC piers experienced almost no damage during severe earthquakes. Two encouraging examples of using SMA-based components for real construction projects have been reported, of which the SMA bars were used in the plastic hinge zones of the RC piers in the State Route 99 Off Ramp Bridge in USA [35], and the SMA-cable-based bearings were installed in the Datianba #2 Bridge in China [36,37]. However, lifecycle assessment of these novel bridge systems, especially their direct, indirect, and long-term economic loss performances, is still insufficient.

With initial confidence gained from the successful practical applications of SMA in bridges, this paper further discussed a novel type of bridge system employing SCR bridge piers, where superelastic shape memory alloy (SMA) washer springs serve as kernel functional components providing self-centering capability and energy dissipation. This new system significantly extends the scope of the practical application of SMA elements in infrastructure. The present study also offers a comprehensive life-cycle assessment framework that evaluates the performance of the new system from both structural and economic perspectives. In the following discussions, the working mechanism of the SCR bridge with the SMA washer-based pier is introduced first. Quasi-static tests on a 1/4 scaled SMA-washer-based RC pier specimen carried out previously by the authors and coworkers are briefly introduced. Subsequently, a performance-based life-cycle assessment flowchart for the SCR bridge system subjected to earthquakes is proposed. Fragility curve, life-cycle loss assessment, and resilience assessment of the SCR bridge are introduced in detail. Finally, a prototype SCR bridge and a conventional bridge are designed and taken as two examples to illustrate the assessment framework. The assessment results, including the fragility curves, life-cycle loss, and resilience performances, are comprehensively discussed.

## 2. Brief Description of Bridge Systems with SMA-Washer-Based Rocking Piers

### 2.1. Working Mechanism and Design Objectives

According to post-earthquake field investigation reports, many bridge systems that followed the displacement-based seismic design philosophy in earthquake-prone areas suffered catastrophic damage or even collapsed. To enhance the resilience of newly designed bridges in earthquake regions, an SMA-washer-based SCR bridge pier was proposed in the previous work by the authors and co-workers [38], as shown in Figure 1. The rocking control capability of the novel bridge system is enabled by its SMA-washer-based SCR piers. The SCR pier is mainly composed of three parts: the upper pile cap, the lower pile cap, and the SMA washer springs (also known as disc springs), which are the kernel components providing self-centering capability and energy dissipation for the bridge system. These washer springs can be stacked either in parallel or in series (or in combination), which makes them flexible in terms of load resistance and available deformability. More technical details of the SMA washer springs can be found elsewhere [39].

In the new pier, each SMA washer spring set consists of several SMA washers, steel bars embedded in the lower pile cap, plastic tubes cast in the concrete of the upper pile cap, and several nuts and shims. As illustrated in Figure 1c, the seismic behavior of such a pier with increasing lateral load could be divided into three stages: (1) decompression stage (Ⅰ) (where the gap is just about to open), (2) post-decompression stage (Ⅱ) (Ⅲ), and (3) locking stage (Ⅳ). In the first stage, the behavior of the pier behaves similar to a fixed pier, i.e., the lateral deformation of the pier relies on the elastic deformation of the pier due to the application of appropriate preload of the SMA washer springs. In the subsequent stage, the lifting force exceeds the decompression force provided by SMA washer spring sets, and the pier starts rocking. When the allowable deformation of washer sets is consumed, the pier is “locked”. Further lateral displacement may rely on the nonlinear deformation of the pier. In light of the above, three basic design goals could be set: (1) the pier does not uplift during small earthquakes; (2) a maximum drift ratio of the bridge is less than the “lock rotation” at moderate (E1) earthquake level; and (3) collapse is prevented at large (E2) earthquake level.

### 2.2. Experimental Verification of SMA-Washer-Based SCR Pier

Figure 2 schematically depicts the test arrangement for the pier specimen. The specimen was held down via four anchor bars passing through the designated slots in the pier base. In order to consider the dead weight of the bridge’s superstructure, a PT tendon was used to apply the axial force. A double-action electro-hydraulic servo actuator was used to provide the lateral load to the loading head, and the lever arm, or the distance between the loading head’s centroid and the rocking interface, was 1625 mm. The RC pier’s diameter and height were 0.3 and 1.05 m, respectively, and other relevant dimensions are shown in Figure 2, with more details given in Fang et al. [38]. The typical test result (hysteretic response) is illustrated in Figure 3. The specimen displayed stable flag-shaped hysteretic curves under cyclic load, with no noticeable decrease in strength and stiffness responses and negligible residual drift. Numerical simulation and system-level analysis of a novel bridge system incorporating the new bridge pier are discussed in detail in Section 4.

## 3. Methodology of Performance-Based Assessment

The main steps of the analysis framework are shown in Figure 4. Seismic fragility analysis and loss assessment are two key steps: the former gives the probabilities of exceeding certain component or system damage levels and the latter enables translation from the damage level to economic loss quantities.

### 3.1. Seismic Fragility Analysis

Structural seismic fragility assessment was carried out first according to the flowchart shown in Figure 4. Fragility analysis is a frequently used technique in the seismic risk assessment in order to calculate the conditional probability of a structure’s or component’s demand reaching or beyond its corresponding capacity [40]. Analytical fragility curves were derived using the probabilistic seismic demand model (PSDM) based on nonlinear time history analysis series. A PSDM is typically developed using two methods: incremental dynamic analysis (IDA) [41] or a cloud technique [42]. The former method requires scaling all the ground motions to specific intensity measurement (*IM*) and conducting a nonlinear time history analysis at each level. In the later procedure, a collection of un-scaled ground motion data is used in the nonlinear time history analysis. Both methods are dependent on *IM*s, and extensive research has been carried out on the selection of suitable *IMs*, considering, for example, Peak ground acceleration (PGA), Peak ground velocity (PGV), and response spectrum type at a specific period. The optimal selection of *IM* may vary with different characteristics of structures [43].

The probabilistic seismic demand model (PSDM) is the probability distribution of structural demand conditioned on specified *IM* and based on the cloud technique. The probability that a structure’s seismic demand (D) exceeds its capacity (C) may be written as follows:(1)$$P[D\geq C|IM]=P[\frac{D}{C}\geq 1]$$

Equation (1) might be rewritten as a lognormal cumulative probability density function provided that C and D have a two-parameter lognormal distribution:(2)$$P[D\geq C|IM]=\Phi (\frac{\ln(S_{d})-\ln(S_{c})}{\sqrt{\beta_{d|IM}^{2}+\beta_{c}^{2}}})$$
where *S_c_* signifies the median structural capacity estimate and *β_c_* denotes the standard deviation. Lognormal median estimate and standard deviation of structural demand in terms of an *IM* are represented by *S_d_* and *β_d_*_|*IM*_, respectively. Regression analysis was used to determine the relationship between *IM* and *S_d_*. The median value of seismic demand, according to Cornell’s power exponent model, may be stated as:(3)$$S_{d}=aIM^{b}\;\mathrm{or}\;\ln(S_{d})=\ln(a)+bln(IM)$$
where *a* and *b* represent the regression parameters obtained from the response analysis. *β_d_*_|*IM*_ can be characterized as:(4)$$\beta_{d|IM}≅\sqrt{\frac{\sum_{i=1}^{n}{\left(ln(d_{i})-ln(S_{d})\right)}^{2}}{N-2}}$$
where *d_i_* represents the structural demand, also known as the seismic response of components, and *i*th represents the earthquake-model sample that corresponds to it. The following steps need to be taken to obtain *β_d_*_|*IM*_:

Given the fragility curves of the components, the fragility curve of a bridge system can be developed according to the first-order reliability theory (explained in Equations (5)–(7)). Such a theory assesses structural performance as an overall system by accounting for the relationship between the vulnerable components. Equation (5) provides the upper and lower bounds of the system fragility functions. The lower bound assumes complete correlation among components, while the upper bound refers to the components with no correlation.
(5)$$\overset{n}{\underset{i=1}{\max}}[P(F_{k})]\leq P(F_{sys})\leq 1-\prod_{i=1}^{n}\left(1-P[F_{component,i}]\right)$$
where *n* is the total number of components that might fail, *P*(*F_k_*) is the probability that the component in concern will fail, *P*[*F_component, i_*] and *P*[*F_sys_*] are the failure probabilities of the *i_th_* component and system, respectively, and Π is the product operator.

If a bridge is supposed to operate as a serial system, with each component executing an essential function separately, any component failure will result in system failure at the same level. As a result, the most significant damage state at the component level is as follows:(6)$$DS_{sys}=\max(DS_{Pier},DS_{Bearing})$$

When a bridge is supposed to be a parallel system, however, it will attain a specific damage state once all of its components have reached that condition. As a result, the system damage state *DS_sys_* is determined by the component with the minimum damage state:(7)$$DS_{sys}=\min(DS_{Pier},DS_{Bearing})$$

The intersection of component probability and its lower and upper limits, as shown in Equation (8), yields the failure probability of a parallel system:(8)$$\prod_{i=1}^{n}P(F_{i})\leq P(F_{sys})\leq \min[P(F_{i})]$$

Independent components are represented by the lower bound, whereas the upper limit represents entirely correlated components. These boundaries are often quite broad, showing the importance of component correlation. In fact, a bridge is neither a parallel nor a serial system, and component responses are often coupled to some degree. The first-order constraints in Equations (5)–(7), which assume total correlation or perfect independence between components, cannot accurately estimate the bridge system’s failure probability. According to the work of Kim et al. [44], the bearing damage due to the load moving to other components of a multi-span simply supported bridge has a substantial impact on the bridge’s overall seismic behavior. The global damage state is hence located in-between the limits set by Equations (5)–(7).

A composite *DS* based on component *DSs*, proposed by Zhang and Huo [45], was employed in this work. Piers and isolation devices were given a weighted ratio of 0.75 and 0.25, respectively, based on their proportional value for load carrying and maintenance cost. This ratio highlights that piers are more important than isolation devices and, as a result, should be given greater weight. However, since either excessive bearing displacement or pier collapse damage (*DS* = 4) might cause a single span or the whole bridge to collapse, a serial mechanism for the collapse damage was used. The following equation summarizes the resulting composite *DS_sys_* for system behavior:(9)$$DS_{sys}=int(0.75\cdot DS_{Pier}+0.25\cdot DS_{Bearing})\;\;\;\;\;\;DS_{Pier},\;DS_{Bearing}<4\;DS_{sys}=4\;DS_{Pier\;or\;DS_{Bearing}\;=\;4}$$

### 3.2. Life-Cycle Loss Assessment

The fragility analysis could be followed by a life-cycle loss assessment, a framework that was initially proposed by the Pacific Earthquake Engineering Research (PEER) Center. Life-cycle loss assessment is an effective tool to evaluate the long-term benefit of the newly proposed bridge system. Direct loss (mostly repair loss) and indirect losses (e.g., running cost and property loss) are important qualities in the life-cycle loss assessment.

The selected seismic events should cover both frequent low-magnitude events with a high probability of occurrence and the high-magnitude earthquakes with a low probability of occurrence. Six hazard events with return periods of 225 years (E1), 475 years (E2), 975 years (E3), 1500 years (E4), 2475 years (E5), and 5000 years (E6) were considered [46]. The relationships between earthquake intensity measurement (*IM*) and the frequency of occurrence for the location of the bridge can be obtained from the USGS national seismic hazard map [47].

The obtained fragility curves were then used to quantify the damage probability of the bridge system. Under a given hazard event, the seismic loss can be calculated by summing up the consequences weighted with the damage probability. Equation (10) gives the expression of the expected annual loss under a specific hazard [48].
(10)$$R=\sum_{LS}C_{LS}P_{LS|IM}$$
where *C_LS_* and *P_LS|IM_* are the consequences at a specific limit state of the bridge and the conditional probability of the bridge at a limit state for a given *IM*, respectively. Direct and indirect losses are the two types of consequences considered in the present study, and the consequences were evaluated in terms of monetary values.

#### 3.2.1. Direct Loss

It is assumed that the necessary repair cost at a certain limit state, *i*, is proportional to the cost needed to rebuild the bridge, as expressed as [49]:(11)$$C_{REP,i}=R_{rcr}\cdot c_{reb}\cdot W\cdot L$$

The total repair cost of a bridge, *C_REP_*_,*i*_, at damage state *i* can be obtained by multiplying the rebuilding cost per square meter (unit: $/m^2^) *c_reb_* by the width *W* and length *L* of the bridge (unit: m), with an extra consideration of the repair cost ratio *R_rcr_* at damage state *i*. As suggested by Mander [50], the repair cost ratios at the slight, moderate, extensive, and collapse levels can be taken as 0.1, 0.3, 0.75, and 1.0, respectively.

#### 3.2.2. Indirect Loss

Societal and economic issues often occur following a seismic hazard, and these consequences result in indirect loss which can be even higher than the direct loss (i.e., repair cost) for highway bridges [51]. The indirect loss after an earthquake is somehow related to structural damage which, for example, affects the traffic flow in the route as the drivers are forced to detour during the closure of the bridge. In this study, the running cost, *C_RUN_*, and the monetary value converted from the time loss for users (i.e., vehicle drivers) through the detour, *C_TL_*, were considered as the indirect loss. *C_RUN_* under a given limit state *i* can be expressed as [49]:(12)$$C_{RUN,i}=[c_{Run,car}(1-\frac{T_{0}}{100})+c_{Run,truck}\frac{T_{0}}{100}]\cdot D_{l}\cdot ADT\cdot d_{i}$$
where *c_Run_*_,*car*_ and *c_Run_*_,*truck*_ are the average running costs for cars and trucks per kilometer ($/km), respectively; *T*_0_ is the average daily truck traffic, defined as the total volume of vehicle traffic of a highway or road for a year divided by 365 days; *D_l_* is the detour length (km); *ADT* is the average daily traffic to detour, which is the average detour distance of all vehicles influenced by the bridge damage. ADT is generally determined by the bridge damage level; *d_i_* is the duration of the downtime associated with the damage levels, where 7, 30, 120, and 400 days are typically adopted corresponding to the slight, moderate, extensive, and complete damage states, respectively [52]. The monetary value of detour-induced time loss, *C_TL_*, can be calculated from:(13)$$C_{TL,i}=[c_{AW}o_{car}(1-\frac{T_{0}}{100})+(c_{ATC}o_{truck}+c_{goods})\frac{T_{0}}{100}]\cdot [\frac{D_{l}\cdot ADT}{S}+ADTE\cdot (\frac{l}{S_{D}}-\frac{l}{S_{0}})]d_{i}$$
where *c_AW_* and *c_ATC_* are the average wage plus compensation per hour ($/h) for car and truck drivers, respectively; *o_car_* and *o_truck_* are the average vehicle occupancies for cars and trucks, respectively; *c_goods_* is the time value of the goods transported in a cargo ($/h); *S* is the average detour speed (km/h); *l* is the route segment containing the bridge (km); *S*_0_ and *S_D_* represent the average speed on the intact link and damaged link (km/h), respectively.

#### 3.2.3. Long-Term Loss

By substituting Equations (11)–(13) into Equation (10), the annual loss of the bridge under a specific hazard event can be calculated. By assuming a Poisson distribution for the occurrence of an earthquake during an investigated time interval (0, *t*_int_), the total life-cycle loss of the bridge can be expressed as [53]:(14)$$LCL_{i}(t_{\mathrm{int}})=\sum_{i=1}^{N(t_{\mathrm{int}})}L_{i}(t_{k})\cdot e^{-\tau t_{k}}$$
where *L_i_*(*t_k_*) is the expected annual loss at time *t_k_*, and *τ* is the monetary discount rate. The total expected lifetime failure loss of the bridge during the time interval, *t*_int_, can be expressed as:(15)$$E[LCL_{i}(t_{\mathrm{int}})]=\frac{\lambda_{f}\cdot E(L_{i})}{\tau }\cdot (1-e^{-\tau t_{\mathrm{int}}})$$
where *λ_f_* denotes the mean rate of the Poisson model. The values of all necessary parameters mentioned above are summarized in Table 1.

### 3.3. Resilience Assessment

Resilience is another important structural performance indicator defining the capability of a civil infrastructure system of maintaining its post-hazard functionality. Generally, resilience includes four gradients, namely, rapidity, robustness, redundancy, and resourcefulness. In this paper, the resilience of the bridges under the considered seismic hazards was assessed, where the functionality is deemed to be resumed when the traffic becomes normal after earthquake.

It is worth mentioning that various functionality levels may be considered for different periods, e.g., emergency response and post-earthquake recovery stages. In the former stage, the main focus should be on the capability of transferring the resources to the disaster area. In the latter phase, the functionality of the bridge can be defined with different service statuses, e.g., “closed”, “limited use”, and “open”. The resilience of the bridge can be evaluated through its recovery pattern. As illustrated graphically in Figure 5, one of the most widely adopted approaches to quantify resilience is [54,55] expressed as follows:(16)$$R_{Resi}=\frac{1}{\Delta t_{r}}\int_{t_{0}}^{t_{0}+\Delta t_{r}}Q(t)dt$$
where *Q*(*t*) is the functionality of a bridge at time *t* (e.g., days); *t*_0_ is the initial time at the investigated point; and *Δt_r_* is the investigated time interval (e.g., days or months).

The shape of the recovery pattern curve is related to the repair and recovery efforts. Generally, bridge functionality can be assessed by defining the damage state to a value between 0 and 1.0, where a value of 0 means collapse of the bridge. Considering various levels of damage state, the bridge functionality can be expressed as:(17)$$Q=\sum_{i=1}^{5}FR_{i}\cdot P_{DS_{i}|IM}$$
where *FR_i_* is the functionality ratio (Func) associated with damage state *i*. For a typical bridge, possible scenarios include immediate access (Func ≥ 0.9), weight restriction (0.6 ≤ Func < 0.9), one lane open only (0.4 ≤ Func < 0.6), emergency access only (0.1 ≤ Func < 0.4), and bridge closure (Func < 0.1). Similar concepts have also been adopted by Padgett and DesRoches [56] and Decò et al. [57]. Once repair actions are initiated, the functionality of the bridge starts to recover and the performance restoration curve starts to rise. The Applied Technology Council (ATC-13) report proposed an approach to quantify the change of functionality *Q_j_*(*t*) during the recovery phase [58], expressed as follows:(18)$$Q_{j}(t)=\frac{1}{\sigma_{j}\sqrt{2\pi }}\int_{-\infty }^{t}\exp[-\frac{{\left(\tau -\mu_{j}\cdot A_{t}\right)}^{2}}{2\sigma_{j}^{2}}]d\tau$$
where *μ_j_* and *σ_j_* are the mean and standard deviation of the recovery time for the *j*th damage state, as listed in Table 2; and *A_t_* is an amplification factor considering the increase of mean recovery time due to underwater repair work. The functionality of the considered bridge on a daily basis during the recovery phase can be calculated with the above-mentioned parameters and methodology. The probabilities of the bridge staying in different damage states can be used to obtain the expected recovery functionality *Q*(*t*), and the resilience can be assessed by Equation (18).

## 4. Case Study

### 4.1. Description of Prototype Bridges

A continuous RC bridge with two equal spans (20 + 20 m) supported by a middle pier was developed to investigate the life-cycle loss and resilience performances of the novel SMA-washer-based SCR bridge system. Figure 6a presents the bridge’s geometric layout. Sliding bearings were placed on each abutment, and fixed bearings were placed on the bent cap above the middle pier. A sufficient separation space between the bridge deck and the abutment was assumed [59,60,61]. The concrete used for the box girder has a compressive strength of 50 MPa, and that for the abutment, pier, and rocking pile caps have a compressive strength of 40 MPa. The longitudinal reinforcement and stirrup utilized in the RC pier have diameters of 32 mm and 16 mm, respectively, with yield strengths of 440 MPa and 300 MPa, respectively.

A total of 55 longitudinal steel bars are equally distributed around the pier perimeter, corresponding to a 1.74% reinforcement ratio. Spiral stirrups are 100 mm apart. The RC pier has a 65-mm-thick concrete cover layer. The rocking pier employed six SMA washer sets, each of which has eight washers, 4 in parallel × 2 in series. Figure 6b shows the key characteristics of each unique SMA washer, where a maximum deformation of 32 mm and a maximum compressive resistance of roughly 250 kN are provided by every single washer. Each SMA washer spring set was preloaded with 960 kN, corresponding to a precompression deformation of 22 mm, to guarantee that the rocking pier does not uplift under normal service loads or small earthquakes. In other words, each SMA washer spring set has remaining deformability of 42 mm prior to the fully compressed status. When the SMA washer sets were totally flattened, a “locking” mechanism was produced, and no further rocking was allowed beyond this allowable angle. After locking, damage to the RC pier is expected.

For comparison, an extra conventional bridge was evaluated with a fixed-base RC pier that is 10 m tall, i.e., equal to the height of the rocking pier from the bottom surface of the bent cap to the rocking interface. The other design of the bridge remains the same.

### 4.2. Numerical Models

Figure 7 shows the behavior of a nonlinear FE model created in OpenSees [62]. Fiber Beam-Column components were used to simulate the pier’s main body, taking nonlinear material features into account. A uniaxial Menegotto–Pinto constitutive model was used to model the behavior of the reinforcement [63,64]. The uniaxial Kent–Scott–Park concrete model was utilized to simulate both the unconfined and confined concrete [65]. The circular portion was separated into eight layers along the radius direction, and each layer was consistently split into 24 fiber components. In addition to the material property, co-rotational geometric transformations were used to account for the geometric nonlinearity [66]. For the pier, the yield curvature was approximately 0.0031 based on the pier’s moment–curvature relationship. By providing an initial strain to the material model of the zero-length element, the preload given to each SMA washer set was considered. As seen in Figure 3, the numerical simulation result of the pier closely matches the actual test results. Rigid beam components were used to simulate the bent cap and the top pile cap. Six pairs of contact point sets were positioned at the interface, and each pair was given a zero-length element to record the change in pressure over time across the rocking interface. Each SMA washer group’s behavior was “lumped” into the zero-length element to account for its overall force-deformation hysteretic behavior. To account for soil–structure interaction (SSI) between the abutment/pile and the soil, a series of zero-length spring components were incorporated [67].

### 4.3. Selection of Ground Motions

A sufficient number of nonlinear time history analyses should be performed to determine the fragility functions. Using the cloud approach, a set of 60 original earthquake data was chosen from the PEER Next Generation Attenuation (NGA) Project ground motion collection. To match the target design spectrum and to offer a relatively wide range of IMs for time history analysis, 60 additional ground motions with a scale factor of 2.0 were included. In other words, this research used a total of 120 ground motions for nonlinear time history analysis. The spectral acceleration at one second (Sa1.0) of all the records, as shown in Figure 8a, has a broad range of values ranging from 0 to 1.6. Figure 8b shows the response spectra of the chosen recordings with the average spectra.

### 4.4. Capacity Models of Bridge Components

The RC pier and bearing are two key components determining the system fragility of the bridges under consideration [68]. In the present research, pier curvature ductility (*μ_f_*) and bearing displacement (*δ_b_*) were regarded as the engineering demand parameters (EDPs), and their peak responses were regarded as the damage indicator. For each EDP, four degrees of damage state were used: slight, moderate, extensive, and complete damage [69], with the damage state treated as a random variable to allow for uncertainty. Based on available test findings, curvature ductility is defined as a range of 0.8 to 7.0 for damage state ranging from slight damage to complete damage [70]. The damage status of the bearings is divided into four levels: 50, 100, 150, and 255 mm bearing displacement. HAZUS proposes a dispersion measure to account for the damage state’s fluctuation, as seen in Table 3.

## 5. Analysis Results and Discussions

### 5.1. Typical Seismic Response of Bridge Components

To take a close look at the pier response, the typical time-history responses of bearing displacement and curvature ductility of the pier are shown in Figure 9a,b, respectively. It could be observed that the introduction of the novel SCR pier bridge system slightly increased the bearing displacement but extensively reduced the pier curvature ductility due to the rocking mechanism. The typical time history responses of the pier’s lateral seismic force versus drift ratio in the conventional and novel bridges are further illustrated in Figure 9c. As anticipated, a flag-shaped hysteretic response with negligible residual deformation was observed for the novel bridge. The SMA washer sets provide moderate energy dissipation. On the other hand, the conventional pier exhibited a fuller hysteresis but was accompanied by damage accumulation. The degree of damage can be further understood by the typical bending moment versus curvature responses at the plastic hinge region of the pier, as shown in Figure 9d. The maximum curvature of the fixed pier was 0.0226, which is significantly larger than that of the rocking pier (i.e., 0.0079). This reaffirms that the fixed pier undergoes more extensive damage than the rocking pier.

### 5.2. Regression Analysis and Optimum IM

When generating the fragility curves for bridges, proper *IM* selection is critical, and much prior research has looked into the optimal *IM* selection for probabilistic seismic risk assessment. The PGA and the spectral acceleration at one second (Sa1.0) are two *IMs* that have been frequently employed in recent research [71,72], although their performance varies depending on the scenario. Both PGA and Sa1.0 were evaluated here. As previously indicated, pier curvature ductility (*μ_f_*) and bearing deformation (*δ_b_*) are the main damage indicators. In a log-transformed space, a linear regression of demand-IM pairs for both *μ_f_* and *δ_b_* was performed (as expressed by ln(*EDP*) = ln*a + b*ln(*IM*), and the results are displayed in Figure 10 and Figure 11.

For optimum *IM* selection, Padgett et al. [71] offered four criteria: efficiency, practicality, proficiency, sufficiency, and hazard computability. Efficiency means less dispersion about the estimated median in the nonlinear time history analysis results and was represented by a lower β. In addition, the coefficient of determination (*R*^2^) was added, which reflects the degree of regression equation fitting. The greater the value of this coefficient, the better the regression. The results showed an evident linear correlation between *EDP* and *IM*, with the result linked with PGA having a more significant degree of dispersion than the result related to Sa1.0. Figure 10 shows, for example, that PSDM’s *R*^2^ values linked with Sa1.0 were more than 0.7, whereas the values related to PGA were less than 0.6. According to the previously established assessment criteria for optimum *IM*, it may be determined that Sa1.0 outperformed PGA in the present investigation.

### 5.3. Fragility Curves

The component fragility curves were directly constructed using Equation (2), which utilizes the capacity models of each limit state, as tabulated in Table 3, and the PSDM parameters determined from the regression analysis. Figure 12 shows the column (pier) and bearing fragility curves for slight, moderate, severe, and total damage states. For slight damage states, the damage probability of bearings associated with conventional and new systems is virtually the same, as shown in Figure 12a. This may be explained by the fact that both types of bridge systems exhibited identical behavior prior to washer spring set decompression. However, since the rocking behavior increases the superstructure displacement, the conditional exceeding probability of the other three bearing damage states for the novel system was somewhat higher than the conventional system. Furthermore, the locking mechanism of the washer spring set helps prevent excessive bearing displacement owing to the controlled rocking behavior. Due to the incorporation of the rocking mechanism, the damage probability of the pier with the novel system was lower than that of the conventional system. The positive impact was more evident considering more severe damage states. For example, assuming Sa1.0 = 0.7 g, the damage probability of a pier with the novel bridge system is 40% lower than that of a conventional bridge for the moderate damage state, and the chances of exceeding extensive and complete pier damage are practically avoided in the novel system.

Figure 13 depicts the system fragility curves associated with the four damage states for both the conventional and novel systems. For slight damage state, the failure probability of the novel bridge system was similar to that of the conventional system, which could be explained by the identical behavior of the two types of systems before decompression of the SMA-washer-based SCR pier. For moderate, severe, and complete damage states, however, it is clear that the seismic performance of the novel system was better than the conventional system, which is due to the mitigated damage to the SMA-washer-based SCR pier and concurrently minor damage to the bearings, according to component fragility curves.

### 5.4. Performance-Based Long-Term Loss Assessment

Following the fragility analysis, an economic loss assessment was carried out, considering the consequences of various damage states. As previously indicated, six hazard scenarios were explored, and the loss associated with these seismic events was calculated. Salt Lake City and Los Angeles are supposed to be the sites of the investigated bridges for illustration purposes. The seismic level of Los Angeles is greater than that of Salt Lake City according to the hazard curve parameters provided by the United States Geological Survey [47]. The Sa1.0 values for Los Angeles is 0.181, 0.279, 0.402, 0.493, 0.604, and 0.791 g, when the return period is 225 years, 475 years, 975 years, 1500 years, and 2475 years, respectively. The Sa1.0 values for Salt Lake City considering the six return periods are 0.070, 0.148, 0.271, 0.362, 0.485, and 0.672 g, respectively. The total direct and indirect cost from all consequences associated with the probability of the bridge being in various damage states were estimated using different hazard scenarios.

Equations (10)–(18) calculate the direct and indirect losses associated with each damage state. The conditional probability of the bridge being in various damage states was calculated directly from the system fragility curve. Figure 14a,b show the total predicted direct and indirect loss from all the six earthquake events examined. Employing the novel bridge system lowers the overall direct and indirect costs, particularly for events 3 through 6. The average decrease rate of the direct loss for these four events was 41.1% in Salt Lake City and 45.7% in Los Angeles. The average decrease rate of indirect loss in Salt Lake City and Los Angeles was 51.2% and 63.0%, respectively. The effectiveness of the SCR bridge system for decreasing the indirect loss was greater than that for decreasing the direct repair loss, as shown in Figure 14b. This is probably because the indirect loss from extensive and complete damage states accounts for a more significant share of the total loss than the direct loss, and the utilization of the SCR bridge system reduces the conditional probability of a bridge system being in a more severe damage state. This also explains why the decrease rate is more significant in locations with higher seismic levels since the investigated bridge has a higher failure chance in such areas. It is also worth noting that when the hazard intensity rises, the indirect loss from social and economic aspects is considerably more significant than the direct loss from repair work, which emphasizes the social and economic consequences of traffic restrictions induced by bridge damage.

In Figure 14c,d, the direct and indirect loss of the conventional and SCR bridge systems under the four damage states are illustrated considering event 5 (i.e., 2475-year return period) and event 6 (i.e., 5000-year return period). It can be found that the loss associated with significant and complete damage states accounted for most of the overall direct or indirect loss for the conventional bridge system. As shown in Figure 14c, the degree of direct loss related to extensive and complete damage remained constant across the conventional and SCR bridge systems. However, the result did not reveal a substantial difference between the two locations under either event 5 or 6. The SCR bridge system reduced the direct loss corresponding to destruction at both sites under events 5 and 6. The primary reason is that using the SCR bridge system minimized the conditional probability of bridge collapse. The SCR bridge system successfully reduces the indirect loss corresponding to all the four damage states to a reasonably low level at the two sites, as illustrated in Figure 14d.

The estimated long-term loss of both the conventional and novel bridge systems is shown in Figure 15a. A life span of 75 years and a monetary discount rate of 2.0% are considered to represent the investigated time period and monetary discount rate respectively. At both sites, the novel bridge system reduced the predicted long-term loss compared with the conventional bridge system. From event 1 through event 6, the long-term loss of the conventional and SCR bridge systems at Salt Lake City kept rising. On the other hand, the long-term loss for Los Angeles peaked in event 5 (i.e., 2475-year return time) and subsequently fell in event 6. From event 3 (i.e., 975-year return period) through event 5 (i.e., 2475-year return period), the peak value of the expected long-term loss of the SCR bridge system remained stable at a relatively low level.

### 5.5. Resilience Assessment

This section evaluates the resilience of the conventional and novel bridges. Resilience is now one of the most important structural performance indicators [73,74,75,76,77,78]. For the conventional bridge system in Los Angeles, the residual functionality for the six analyzed events was 0.95, 0.76, 0.46, 0.30, 0.18, and 0.09, respectively. In contrast, the residual functionality for the SCR bridge system was 0.95, 0.84, 0.66, 0.56, 0.46, and 0.31, respectively. As anticipated, an increasing seismic intensity leads to decreased residual functionality, and the SCR bridge successfully increases the residual functionality more than the conventional system. The residual functionality of the conventional bridge system at Salt Lake City was 1.00, 0.98, 0.78, 0.56, 0.31, and 0.14, respectively, whereas the novel system’s values were 1.00, 0.98, 0.86, 0.71, 0.57, and 0.41, which, again, were much increased.

The resilience of the bridge was further assessed using Equation (16) and through the functionality restoration function, i.e., Equation (18). Figure 16a,b depict the functionality of the damaged bridge systems after recovery procedures, considering events 5 (i.e., 2475-year return period) and 6 (i.e., 5000-year return period), respectively. It can be seen that when the recovery effort was initiated, the predicted functionality of the damaged bridge systems grew with time. Importantly, the SCR bridge system outperformed the conventional system in terms of recovery time. As shown in Figure 16a, the predicted functionality of the SCR bridge system in Los Angeles returned to 0.9, i.e., rapid access, after 110 days following the occurrence of an earthquake under event 5. In contrast, the conventional bridge system took 290 days. A similar pattern is seen in Figure 16b. This phenomenon may be explained by the fact that the indirect loss of the SCR bridge system is substantially lower than that of the conventional system. In particular, quick recovery after an earthquake may drastically decrease the social and economic losses. As previously stated, the SCR bridge system considerably improved the resilience, and the improvement is more significant with increased hazard intensity. In addition, the resilience enhancement in a more seismically active location, such as Los Angeles, is more substantial.

## 6. Conclusions

This paper presented a performance-based framework for evaluating the life-cycle loss and resilience of a conventional bridge and a novel SCR bridge with SMA washer-based piers, and successfully demonstrated the benefit associated with the adoption of this novel pier. Fragility curves of the conventional and SCR bridge systems were obtained by conducting nonlinear time-history analysis. Life-cycle loss and resilience under the investigated hazard scenarios were assessed, considering the direct and indirect loss and hazard recovery pattern. The proposed assessment framework was applied to continuous RC bridges with and without incorporating the SCR pier. This work provided an efficient decision-making tool for the application of new bridge systems in the structural seismic design process. The following conclusions can be obtained.

The introduction of the novel SCR pier bridge system slightly increased the bearing displacement but extensively reduced the pier curvature ductility due to the rocking mechanism; the damage probability of the pier with the novel system was lower than that of the conventional system. The positive impact was more evident considering more severe damage states.The long-term loss of the conventional and SCR bridge systems within the investigated time was assessed and compared. The indirect loss increased with a higher hazard intensity. Indirect loss is much larger than the direct loss, specifically under the earthquakes with a relatively low probability of occurrence. As can be concluded from the results, the life-cycle loss of the bridge using the SCR bridge piers can be reduced significantly. The investigated time interval can affect the life-cycle loss remarkably.The performance benefit of resilience associated with the SCR bridge system increase with a larger investigated hazard intensity. The SCR bridge system is more beneficial for the bridges located in seismic zones with higher hazard intensity.The investigated seismic intensity can affect the resilience significantly. The SCR bridge system outperformed the conventional system in terms of recovery time. Quick recovery after an earthquake may drastically decrease the social and economic losses. The difference of the resilience performance between the conventional and SCR bridge systems increased with the increase of the investigated hazard intensity.

## Figures and Tables

**Figure 1 materials-15-06589-f001:**
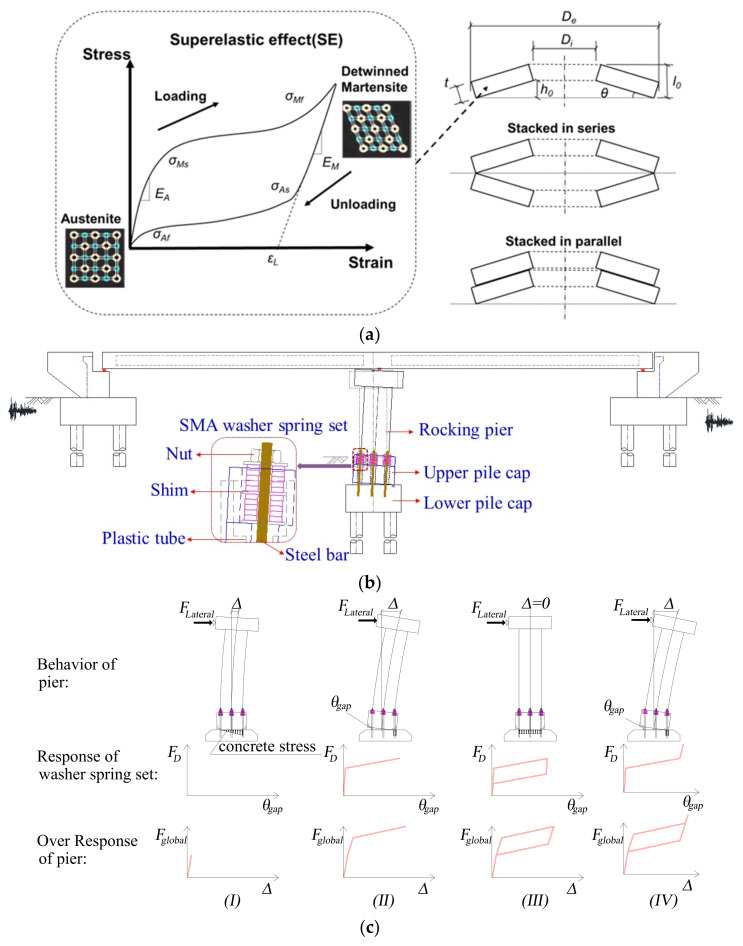
SMA-washer-based SCR bridge pier: (**a**) illustration of the SMA washer springs, (**b**) illustration of the bridge system, (**c**) working principle of the SCR bridge pier.

**Figure 2 materials-15-06589-f002:**
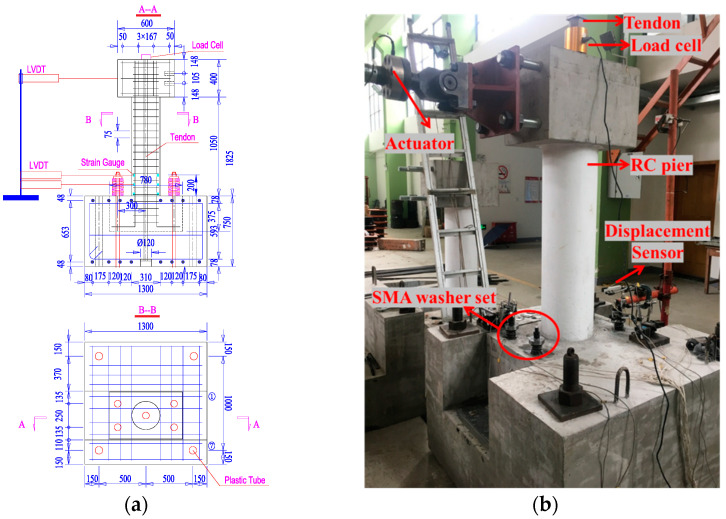
Test of the SCR pier: (**a**) drawing of the SCR pier specimen, (**b**) photo of the test setup.

**Figure 3 materials-15-06589-f003:**
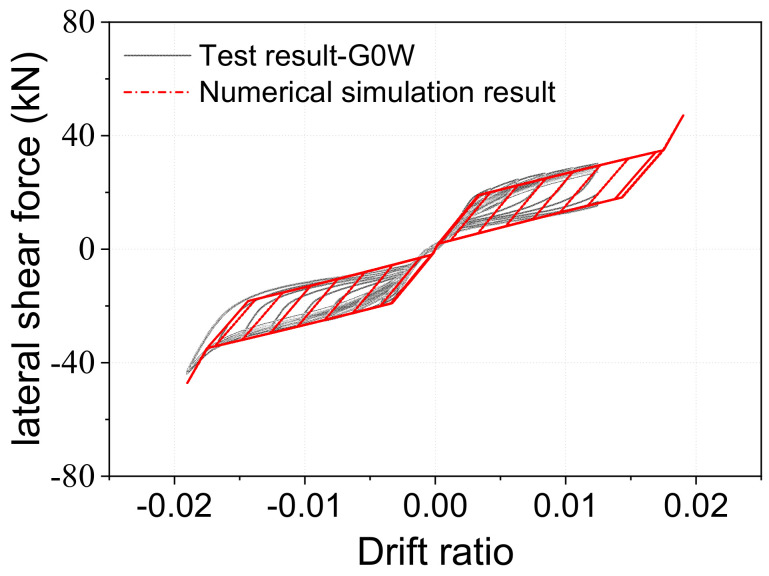
Shear force–drift ratio hysteretic curves of the test and numerical simulation results.

**Figure 4 materials-15-06589-f004:**
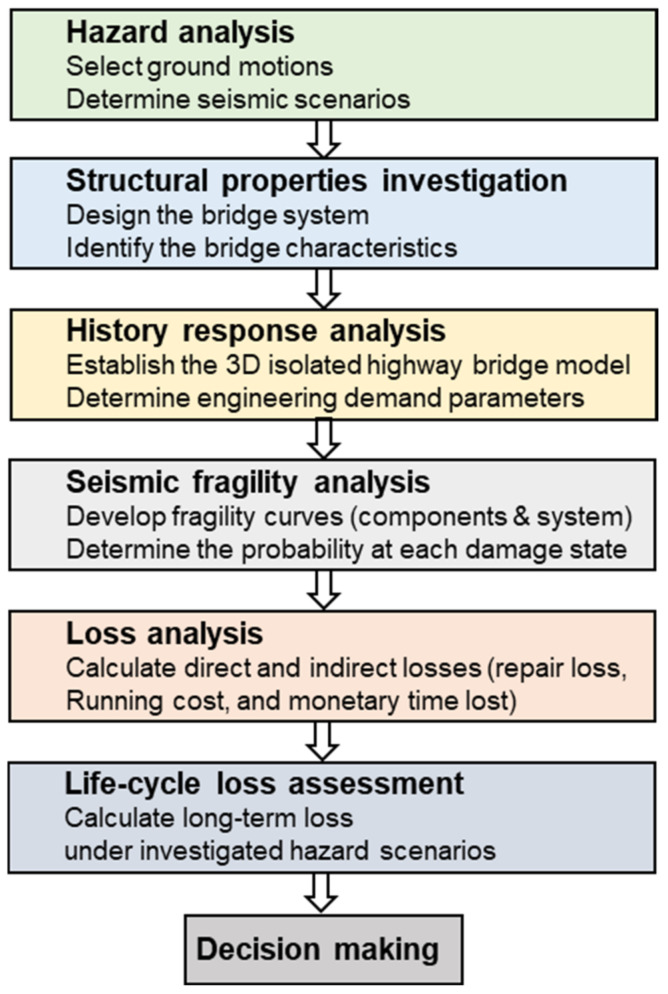
Flowchart of the performance-based life-cycle assessment of bridge systems under seismic hazard.

**Figure 5 materials-15-06589-f005:**
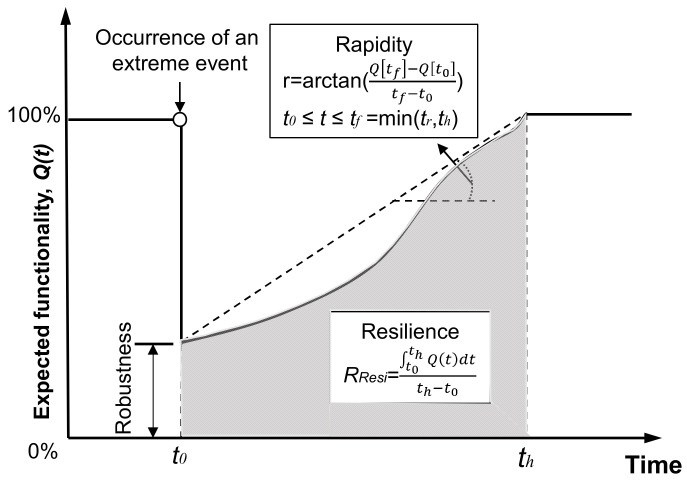
Schematic representation of resilience.

**Figure 6 materials-15-06589-f006:**
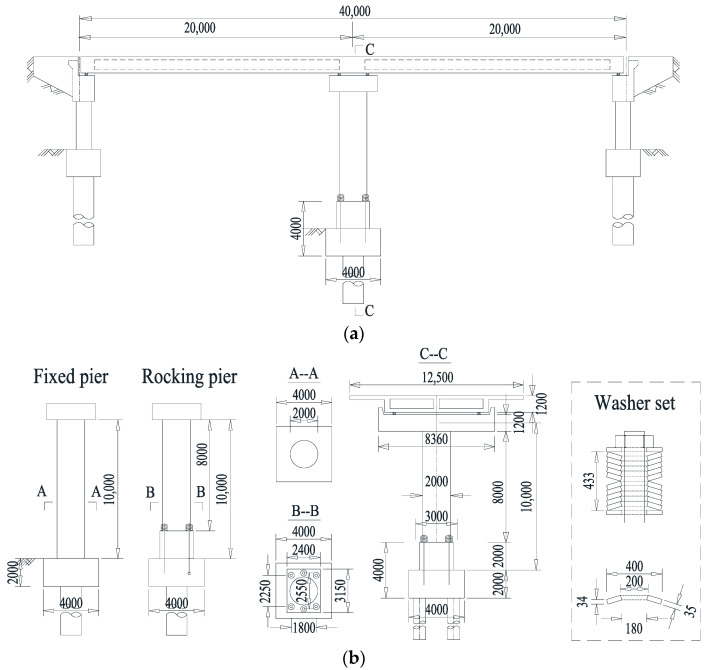
Geometric layout of two-span continuous concrete-girder bridge (Unit: mm): (**a**) general layout, (**b**) detailed dimensions.

**Figure 7 materials-15-06589-f007:**
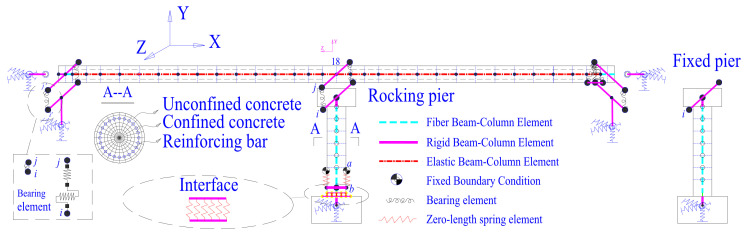
Schematic illustration of the FE bridge model and modeling details.

**Figure 8 materials-15-06589-f008:**
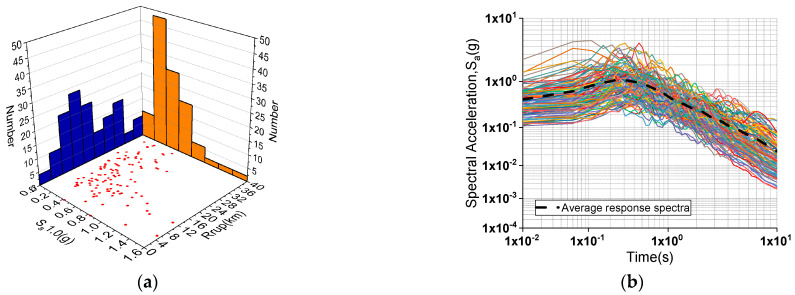
Information of selected earthquake records: (**a**) Sa1.0 and fault distance; (**b**) spectral acceleration.

**Figure 9 materials-15-06589-f009:**
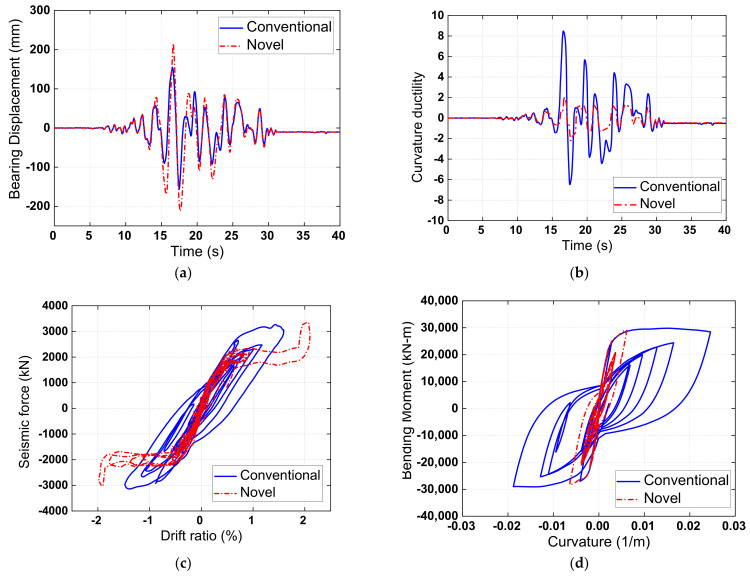
Seismic response of (**a**) bearing displacement, (**b**) curvature ductility, (**c**) seismic force versus drift ratio behavior, and (**d**) curvature versus bending moment behavior.

**Figure 10 materials-15-06589-f010:**
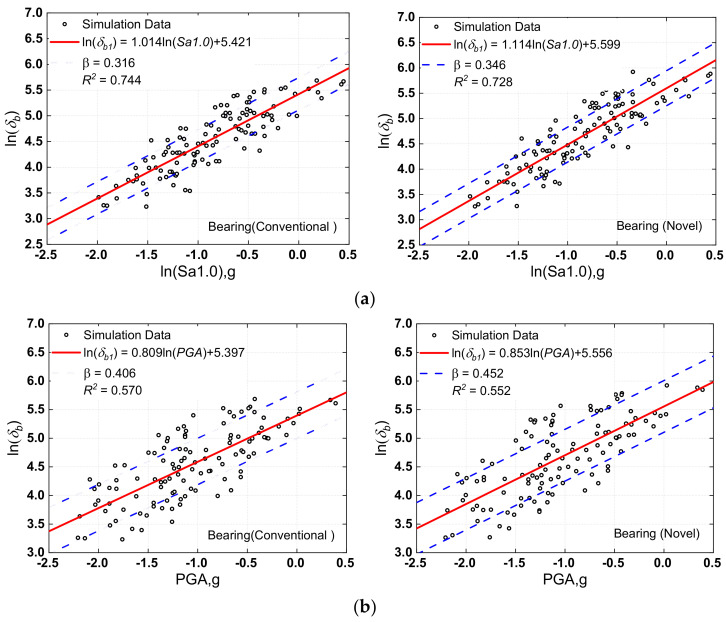
Probabilistic seismic demand models of bearing: (**a**) Sa1.0 versus bearing displacement responses, (**b**) PGA versus bearing displacement responses.

**Figure 11 materials-15-06589-f011:**
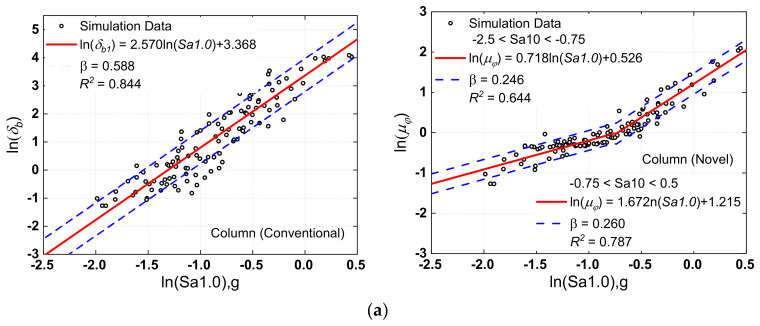
Probabilistic seismic demand models of column (pier): (**a**) Sa1.0 versus column curvature ductility responses, (**b**) PGA versus column curvature ductility responses.

**Figure 12 materials-15-06589-f012:**
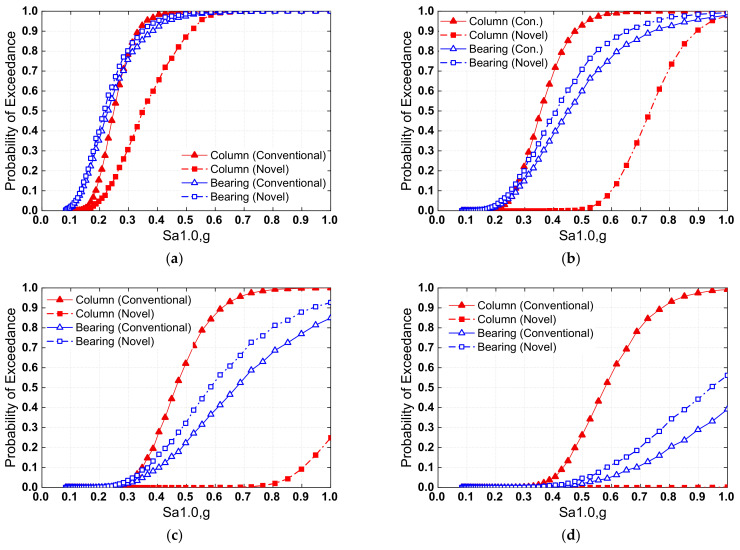
Component fragility for (**a**) slight damage, (**b**) moderate damage, (**c**) extensive damage, and (**d**) complete damage.

**Figure 13 materials-15-06589-f013:**
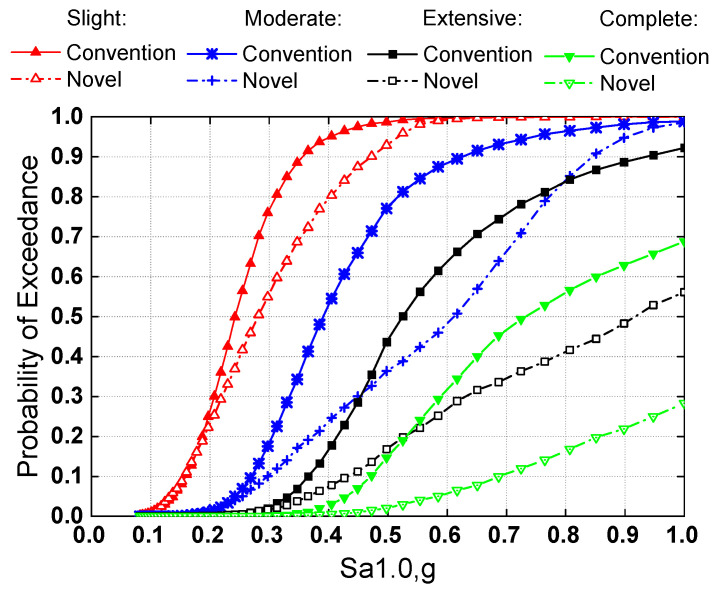
System fragility curve of conventional and novel bridges.

**Figure 14 materials-15-06589-f014:**
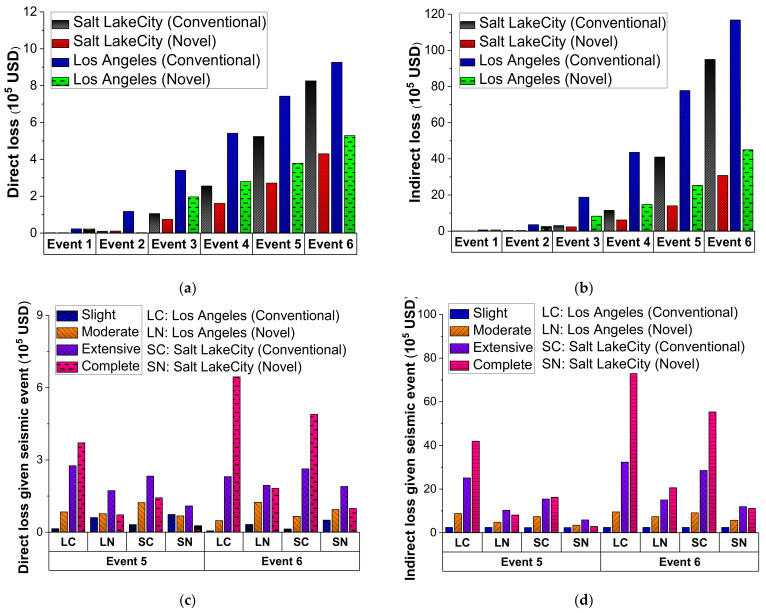
Predicted losses of bridges under different hazard events: (**a**) total direct loss, (**b**) total indirect loss, (**c**) direct loss under four damage states, and (**d**) indirect loss under four damage states.

**Figure 15 materials-15-06589-f015:**
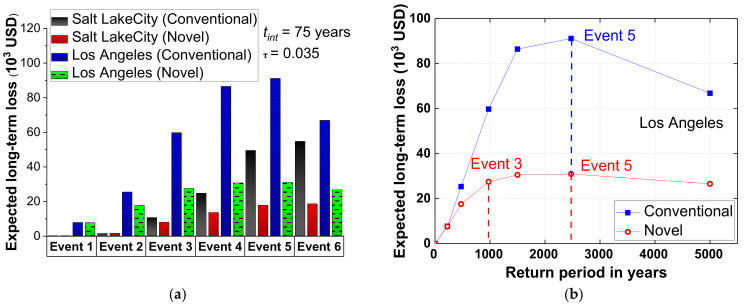
Expected long-term loss of bridges (**a**) under six events (**b**) with 50, 475, 2475, and 5000-year return periods.

**Figure 16 materials-15-06589-f016:**
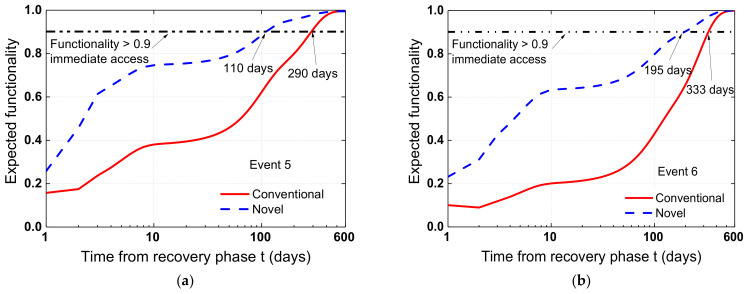
Expected functionality of the bridge from the recovery phase under (**a**) event 5 and (**b**) event 6.

**Table 1 materials-15-06589-t001:** Parameters associated with consequences.

Parameters	Notation	Value	References
Average daily truck traffic	*ADTT*	19,750	[51]
Average daily traffic on the damage link to average daily traffic	*ADTE/ADT*	0.12	[51]
Daily truck traffic ratio	*T_0_*	13%	[53]
Link length (km)	*l*	6	[53]
Detour additional distance (km)	*D_1_*	2	[53]
Vehicle occupancies for cars	*o_car_*	1.5	[49,53]
Vehicle occupancies for trucks	*o_truck_*	1.05	[49,53]
Wage for car drivers ($/h)	*c_AW_*	11.91	[49,53]
Compensation for truck drivers ($/h)	*c_ATC_*	29.87	[49,53]
Operating costs for cars ($/km)	*c_Run,car_*	0.4	[49,53]
Operating costs for trucks ($/km)	*c_Run,truck_*	0.57	[49,53]
Rebuilding costs ($/m^2^)	*c_reb_*	2306	[50]
Detour speed (km/h)	*S*	50	[53]
Link speed (km/h)	*S_0_*	80	[53]
Time value of a cargo ($/h)	*c_goods_*	4	[53]
Monetary discount rate	*τ*	2%	[53]

**Table 2 materials-15-06589-t002:** Parameters associated with bridge restoration functionality [58].

Damage State	Mean (Days)				Coefficient of Variation
	Lower Limit	Upper Limit	Mode	Distribution Type	
Slight	0.2	1	0.6	Triangular	1
Moderate	1	5	2.5	Triangular	1
Major	30	120	75	Triangular	0.56
Complete	120	360	230	Triangular	0.48

**Table 3 materials-15-06589-t003:** Engineering demand parameters and damage states.

	*DS*_1_ (Slight)		*DS*_2_ (Moderate)		*DS*_3_ (Extensive)		*DS*_4_ (Complete)	
Component	S_c_	β_c_	S_c_	β_c_	S_c_	β_c_	S_c_	β_c_
**Column curvature ductility (*μ_f_*)**	0.8	0.005	2	0.005	4	0.005	7	0.005
**Bearing displacement(mm)**	50	0.25	100	0.25	150	0.46	255	0.46

## Data Availability

Data presented in this study are available in this article.

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
