# Peer review of "Performance-Based Assessment of Bridges with Novel SMA-Washer-Based Self-Centering Rocking Piers"

_materials, 2022, doi:10.3390/ma15196589_

Round 1

Reviewer 1 Report

The paper titled “Performance-based assessment of bridges with novel SMA-washer-based self-centering rocking piers” reports a novel type of bridge system employing SCR bridge piers, where superelastic shape memory alloy (SMA) washer springs serve as kernel functional components providing self-centering capability and energy dissipation. The paper presents new and very important information, which is believed to be attractive for the researchers and the bridge builders. I recommend this paper for publication after minor revision. The comments are listed below.

1.      Page 2: “A representative investigation was conducted by Sebastian and Saiidi [20],” – Actually, Sebastian is the given name. The correct reference is Varela, S.; Saiidi, MS. And the sentence should be “A representative investigation was conducted by Varela and Saiidi [20],”

2.      Page 3: “at large earthquake (E2) earthquake level” -?-> “at large earthquake (E2) level”

3.      Figure 1 caption: “(a) information of SMA washer springs” -?-> “(a) illustration of SMA washer springs”

4.      Figure 2 caption: “Information of the SCR pier specimen and test setup [38]” -> “Drawing of the SCR pier specimen (a) and photo of the test setup (b) [38]”

5.      Eq. 18: Is it μ or μj?

6.      Table 2: The Mode parameter is not explained.

7.      The equations of the fitting lines in Figs. 10 (b) and 11 (b) includes ln(Sa10). Is it not a mistake?

Author Response

Responses can be seen in the attached file.

Reviewer 2 Report

The paper deals with a novel self-centering rocking (SCR) bridge system equipped with shape memory alloy (SMA)-based piers, with a particular focus on the benefit of the SCR bridge system in a life-cycle context.

At first, the methodology adopted is introduced and commented. Then, an application to case study is presented in order to evaluate the benefits in a life-cycle context.

It is opinion of the Reviewer that the paper is well structured and deserves to be considered for the publication the following reasons:

- the contents are clearly described;

- there is a clear description of the methodology adopted;

- an application to a case study is shown and commented. 

Author Response

(The authors gave the same response as above.)

Reviewer 3 Report

The present manuscript presents a new type of bridge system design. It is a self-centering rocking (SCR) bridge system. The paper is scientific in nature and is written in a clear and understandable format. This manuscript has no fundamental flaws. The research methodology, objectives, discussion and conclusions are adequate for a scientific article. The article contains analytical and experiemental approach. The only thing I would recommend to add in the future is the numerical modeling of the bridge structure. This method allows to take into account several phenomena and to approximate the model to reality.

Author Response

(The authors gave the same response as above.)
